# *Query of CC*: Unearthing Large Scale Domain-Specific Knowledge from Public Corpora

**Zhaoye Fei[1,2♦], Yunfan Shao[1,2♦], Linyang Li[1,2♦], Zhiyuan Zeng[1,2]**
**Conghui He[2], Hang Yan[2], Dahua Lin[2] and Xipeng Qiu[1]**
[1]School of Computer Science, Fudan University, Shanghai, China
[2]Shanghai AI Laboratory
{zyfei20, yfshao19, xpqiu}@fudan.edu.cn
{lilinyang, yanhang}@pjlab.org.cn

## Abstract

Large language models (LLMs) have demonstrated remarkable potential in various tasks, however, there remains a significant lack of open-source models and data for specific domains. Previous work has primarily focused on manually specifying resources and collecting high-quality data for specific domains, which is extremely time-consuming and labor-intensive. To address this limitation, we introduce large models into the data collection pipeline to guide the generation of domain-specific information and retrieve relevant data from Common Crawl (CC), a large public corpus. We called this method as *Query of CC*. It not only collects data related to domain-specific knowledge but also mines the data with potential reasoning procedures from the public corpus. By applying this method, we have collected a knowledge domain-related dataset named KNOWLEDGE PILE, which covers four main domains, including the sciences, humanities, and other categories. Through the analysis of KNOWLEDGE PILE, *Query of CC* can effectively retrieve relevant data from the covered knowledge domains and significantly enhance the performance in tests of mathematical and knowledge-related reasoning abilities. We have open-sourced our data on HuggingFace to promote academic progress in knowledge reasoning capabilities.

## 1   Introduction

Large language models (LLMs) are becoming the new trend not only in natural language processing but also in the entire AI community, pioneered by OpenAI ChatGPT and GPT-4 [OpenAI, 2023]. While commercial LLMs are close-sourced, open-source models such as LLaMA [Touvron et al., 2023a] and Mistral [Jiang et al., 2023] are widely studied by the community since they serve as general base models for building LLM applications. Based on these base models, domain-specific models, show great potential in specific domains, such as medicine [Yang et al., 2022, Gao et al., 2023], finance [Wu et al., 2023, Zhang and Yang, 2023], science [Taylor et al., 2022a, Wei et al., 2023], and law [Nguyen, 2023, Cui et al., 2023]. These domain-specific enhanced models are based on specific human-crafted data recipes [Azerbayev et al., 2023, Wang et al., 2023a].

However, crafting domain-specific data is very costly. As depicted in Figure 1a, traditional data collection methods involve the selection of relevant resources by domain experts, followed by data

---

[*] Work done during internship at Shanghai AI Laboratory.

[♦] These authors contributed equally to this work.

[1] KNOWLEDGE PILE will be released in `https://huggingface.co/datasets/Query-of-CC/knowledge_pile_full`.

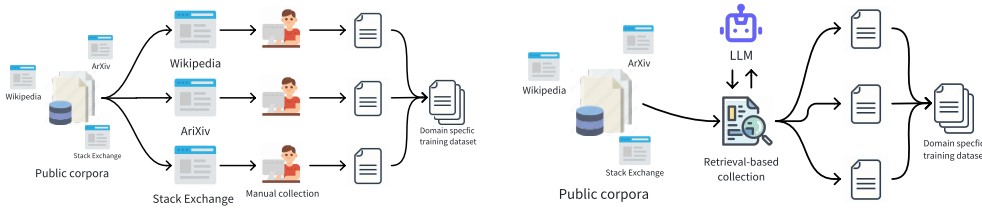

(a) Manual Data Collection       (b) Query and Retrieve Data Collection(our approach)

Figure 1: Comparison of traditional manual data collection methods with our approach.

collection and processing by engineers. On the one hand, such endeavors are highly labor-intensive, requiring several months of collaboration between multiple domain experts and engineers for corpus collection. On the other hand, some specific domain-related data distribution may be highly scattered, which poses many challenges for large-scale domain-specific data collection. Therefore, in this paper, we introduce an automatic strategy to retrieve data from public corpora for specific domain knowledge bootstrapping, which we call *Query of CC*.

In *Query of CC*, we initially collected seed information in some specific domains, such as keywords, frequently asked questions, and textbooks, to serve as inputs for the Query Bootstrapping stage. Leveraging the great generalization capability of LLMs, we can effortlessly expand the initial seed information and extend it to an amount of domain-relevant queries. Inspiration from Wang et al. [2023b] and [Xu et al., 2023], we encompassed two stages of expansion, namely **Question Extension** and **Thought Generation**, which respectively extend the queries in terms of breadth and depth, for retrieving the domain-related data with a broader scope and deeper thought. Subsequently, based on the queries, we retrieved relevant documents from public corpora, and after performing operations such as duplicate data removal and filtering, we formed the final training dataset.

Otherwise, leveraging *Query of CC*, we collect a high-quality knowledge dataset KNOWLEDGE PILE, which starts from some seed information of four major domains, including STEM, humanities, social sciences, and medical sciences, as well as general knowledge. Utilizing KNOWLEDGE PILE, we enhance the Llama and Mistral models through continuing learning. Experimental results indicate that through the KNOWLEDGE PILE, both Llama and Mistral enhanced models achieved significant performance improvements over baselines in benchmark tests related to mathematics, knowledge assessments, professional examinations, and some complex reasoning tasks.

To sum up our contribution:

- We propose *Query of CC*, a data collection pipeline to retrieve domain-specific knowledge from public corpora, which introduces LLMs to extend query and retrieve domain-related data from public corpora.

- We collect and release a knowledge-related corpora KNOWLEDGE PILE based on *Query of CC* from Common Crawl, a large-scale public corpora, which includes various categories such as STEM, human science, and social science.

- We have analyzed for the quality and statistical of KNOWLEDGE PILE. we statistic the distribution of web domain to show the performance of *Query of CC* in collecting scattered information, then, we compared the educational value of KNOWLEDGE PILE with other open source knowledge-related datasets, to show the high educational value of our dataset.

- We train several language models on KNOWLEDGE PILE, which demonstrate significant improvements on several professional exams and reasoning datasets.

## 2 Related work

### 2.1 Large language model for knowledge-based reasoning

In recent years, significant progress has been made in the field of Natural Language Processing (NLP), driven by the emergence of large language models [OpenAI, 2023, InternLM-Team, 2023, Bai

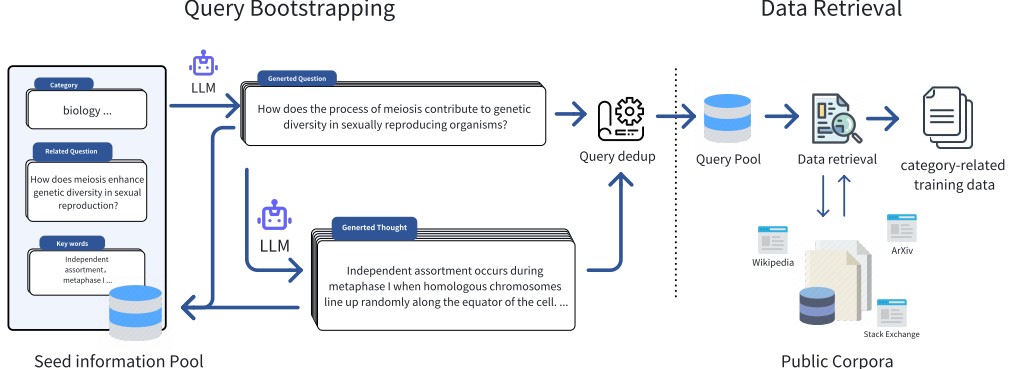

Figure 2: The overview of *Query of CC*'s two major components: Query Bootstrapping and Data Retrieval.

et al., 2023, Sun et al., 2023]. Particularly, in the domain of academic and professional examinations, Some language models such as ChatGPT and GPT-4 [OpenAI, 2023] have demonstrated remarkable success in solving complex tasks, achieving human-like performance through the utilization of the capability of reasoning [Wei et al., 2022, Wang et al., 2023c]. However, open-source LLMs lag in performance (Like Llama [Touvron et al., 2023a], Mistral [Jiang et al., 2023] etc.), possibly due to a lack of data.

## 2.2 Manual Data Collection

Currently, extensive efforts are being dedicated to the manual collection of specific training data to enhance the capabilities of Large Language Models (LLM) in knowledge-based reasoning. In the field of mathematics, Lewkowycz et al. [2022] undertook the task of gathering approximately 40 billion tokens of data from arXiv and web math pages. They developed a series of Minerva models based on PaLM [Chowdhery et al., 2023] and observed that augmenting the model with more mathematical data significantly enhances its proficiency in mathematical reasoning. Similarly, numerous works [Azerbayev et al., 2023, Wang et al., 2023a, Paster et al., 2023] have undertaken the collection of mathematics-related data, encompassing papers, web pages, and code, with considerable cost.

In the academic and technological domains, Taylor et al. [2022b] collected 106 billion academic and technological data. They asserted that the resulting 120B Galactica model surpasses GPT-3 in various academic benchmarks. These works highlight the efficacy of manual data collection in enhancing model performance. However, it is crucial to note that these data collection endeavors are labor-intensive, and present challenges in scalability, thereby posing some constraints on the overall improvement of model performance.

In contrast to these human-centric collection methods, our approach delves into an automated method for bootstrapping and collecting domain-specific data from public corpora and aims to achieve scalability, efficiency, and cost-effectiveness in data collection.

## 2.3 Retrieval-based Data Collection

Many works [Li et al., 2023, Li and Qiu, 2023] utilized retrieval methods to enhance their capabilities. The majority of these focus on retrieving documents relevant to the questions to improve the model's prior knowledge, thereby enhancing the performance on knowledge-related tasks and reducing hallucinations. Also, [Yue et al., 2024] retrieved related data for enhancing the instruction synthetic. For data collection, some works [Yao et al., 2022] attempt to use retrieval during the training phase for data collection to improve specified downstream tasks. However, Retrieving specified information for specific downstream tasks relies on the data of those tasks, making it difficult to automate and scale. In contrast, our method introduces LLMs to automatically extend domain-related queries, which enhances the automation and scalability of data collection.

# 3 *Query of CC*

## 3.1 Overview

The overview of *Query of CC* is illustrated in Figure 2. This framework encompasses two major stages, namely **Query Bootstrapping** and **Data Retrieval**. During the Query Bootstrapping phase, the Large Language Model constructs questions and answers centered around the given keywords to retrieve relevant text from public corpora. Through this bootstrapping process, we construct queries around the seed keywords in both depth and breadth, ensuring that the query encompasses a wide range of knowledge and sufficient depth. In the Data Retrieval phase, we employed the BM25 algorithm to retrieve text relevant to specific categories and obtained training data through post-processing steps such as deduplication.

## 3.2 Query Bootstrapping

To efficiently and comprehensively retrieve high-quality data relevant to the given seed information, we employed the **Query Bootstrapping** method, inspiration from Wang et al. [2023b]. During this phase, we expanded queries around seed information such as category or some related questions, by querying large language models and aims to extend the scope of retrieved data from both breadth and depth perspectives.

**Question Extension** To broaden the scope of our queries, we leverage large language models to generate questions relevant to the given seed words which we called **Question Extension**. Leveraging the capabilities of large models for question generation is evident in expanding the conceptual boundaries of the keywords. This process evolves a narrow category into a more comprehensive representation, encompassing various concepts associated with the given words. Consequently, it significantly enhances the breadth of the query set, ensuring a more comprehensive coverage of various aspects within the target domain.

**Thought Generation** In addition to expanding the range of queries through Question Evolution, we are dedicated to enhancing the depth of thought in generated queries through **Thought Generation**. In this phase, we are inspired by the work of Wei et al. [2022], employing LLMs to generate the cognitive processes necessary for answering questions. This enables us to acquire detailed and insightful responses. This approach supports a more thorough exploration of concepts related to seed information and generate the cognitive processes essential for answering questions. The generated thoughts not only serve for retrieving seed information-related data but also, by expanding queries, offer additional context and relevant knowledge, contributing to a more comprehensive understanding of the subject matter.

**Query post processing** We conducted post-processing operations on queries, involving two main stages: pre-cleaning before entering the seed information pool and cleaning and filtering before entering the query pool. On one hand, the generated questions underwent a re-cleaning process and were incorporated into the seed information pool for the next rounds of question bootstrapping and thought generation. This cleaning step involved removing incompletely generated language data to prevent the influence of non-natural language. On the other hand, before entering the query pool, we employed Minhash-LSH [Broder, 1997] deduplication on queries to mitigate performance inefficiencies resulting from redundant retrievals.

## 3.3 Data Retrieval

Based on the query bootstrapping stage, we get extensive and in-depth queries. During the data retrieval stage, utilizing the enriched queries, we employ the BM25 [Robertson and Walker, 1994] algorithm to retrieve data from general public corpora. BM25 is a widely adopted relevance calculation method commonly used by search engines. It calculates the relevance score between the given query and target documents by weighting and summing the matching degree of keywords in the query with the target documents. Efficiency is the reason why we use BM25 to calculate the relevance. When dealing with billion data, performing relevance calculations for each query against every document becomes exceedingly challenging, while BM25 rapidly retrieves documents relevant to the target query. Compared with Dense Retriever [Karpukhin et al., 2020], it may incur a potential loss in

| Datasets | Target domain | automatic | methods | source | Data scale | Open Source |
|---|---|---|---|---|---|---|
| Proof of Pile | Mathematics reasoning | ✗ | human collection | arXiv, Textbooks, Lib., Stack Exchange, ProofWiki, MATH | 8.3B | ✓ |
| OpenWebMath | Mathematics reasoning | ✗ | human collection | Common Crawl | 14.7B | ✓ |
| MATHPILE | Mathematics reasoning | ✗ | human collection | arXiv, Textbooks, Lib., Stack Exchange, ProofWiki, MATH, Web | 9.5B | ✓ |
| AcademicGPT | Academic | ✗ | human collection | arXiv, Unpaywall, Top Universities, Pubmed, Common Crawl, Semantic Scholar, Wiki | 370B | ✗ |
| Galactica | Academic | ✗ | human collection | Papers, Code, Reference Material, Knowledge Bases, Common Crawl, Prompts | 106B | ✗ |
| KNOWLEDGE PILE (ours) | Mathematics reasoning & Knowledge | ✓ | automatic query and retrieve | Public Corpora | 188B | ✓ |

Table 1: Comparison of KNOWLEDGE PILE with other specific domain knowledge dataset. In this table, most data scales are derived from publicly released research papers, while the data scale for the KNOWLEDGE PILE is obtained through tokenized data analysis using the Llama2 tokenizer.

retrieval accuracy, but the latter comes with an unbearable high computational cost. Exploring the potential impact of retriever selection on the quality of collected data might be a valuable direction for future research.

For each query $q_i$, we conduct the relevance score against every document $d$ in the public corpora $\mathcal{D}$. Followed by sorting the documents based on relevance, we retrieve top-k document set $\mathcal{S}_i = \{\tilde{d}_1, ..., \tilde{d}_k\}$ with the highest relevance with the query $q_i$. In our experiments, the typical choice for $k$ is 1000. The retrieved data which related all the query $q_i \in \mathcal{Q}$ is consolidated into training dataset $\mathcal{S} = \cup_i \mathcal{S}_i$.

# 4 KNOWLEDGE PILE

Leveraging *Query of CC*, Based on queries of some knowledge categories, we retrieved several knowledge-related data from processed public corpora. We call the collected datasets as KNOWLEDGE PILE. In this section, we will introduce the analysis of queries (Section 4.1) and the analysis of KNOWLEDGE PILE (Section 4.2). Also, we train several language models to show the improvement of KNOWLEDGE PILE in some knowledge-related reasoning benchmarks (Section 4.3). Otherwise, we discuss about the different when improving different language models using KNOWLEDGE PILE (Section 4.4). See more implement details in Appendix.

## 4.1 Query Analysis

The progress of query bootstrapping initiates from some categories. Inspiration of the classification of Hendrycks et al. [2021], we select multiple categories for our initial seed information in the STEM (Science, technology, engineering, and mathematics), Humanities sciences, Social Sciences, and miscellaneous. The key keywords for each category are as follows:

**STEM**: mathematics, physics, chemistry, biology, computer science, engine;

**Humanities**: logical, history, law, philosophy, religions;

**Social science**: econometrics, politics, psychology, sexuality, public relations, psychology, sociology;

**Misc**: medicine, virology, commonsense knowledge and other miscellaneous.

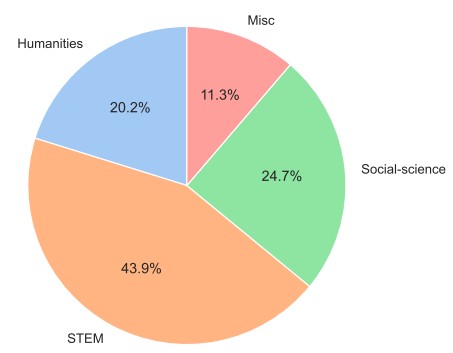

Figure 3: The category distribution of the query for KNOWLEDGE PILE.

After multiple rounds of iterative augmentation and deduplication, we obtain a total of 340,000 queries. The distribution of queries across different domains is depicted in Figure 3. In our query pool, STEM-

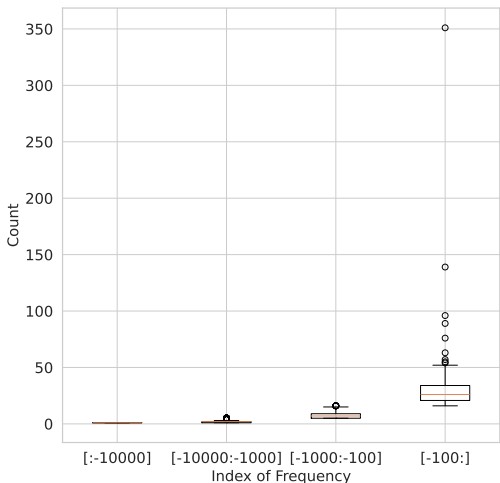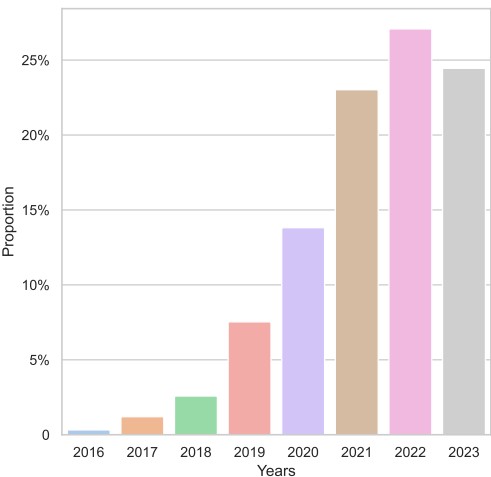

Figure 4: Left: The count distribution of index of web domain frequency. *Query of CC* not only retrieve the data from high knowledge density websites like Wikipedia, but collect data from scatted websites. Right: The timestamp statistics of KNOWLEDGE PILE, most data of KNOWLEDGE PILE come from recent years.

related queries constitute the majority, while the proportion of queries similar to miscellaneous is relatively small.

## 4.2 Data Analysis

**Overview**  Based on *Query of CC*, we have formed a high-quality knowledge dataset KNOWLEDGE PILE, which maintains about 735GB disk and 188B tokens (using Llama2 tokenizer). As shown in Figure 1, comparing with other datasets in academic and mathematical reasoning domains, we have acquired a large-scale, knowledge-related dataset at a lower cost, without the need for manual intervention. Through automated query bootstrapping, we efficiently capture the information about the seed query. KNOWLEDGE PILE not only covers mathematical reasoning data but also encompasses rich knowledge-oriented corpora spanning various fields such as biology, physics, etc., enhancing its comprehensive research and application potential.

**Web Domain composition of *Query of CC***  Table 2 presents the top 10 web domains with the highest proportion in KNOWLEDGE PILE, which cover a wide range of academic institutions, high-value fo-

| Web Domain | Count |
|---|---|
| en.wikipedia.org | 398833 |
| www.semanticscholar.org | 141268 |
| slideplayer.com | 108177 |
| www.ncbi.nlm.nih.gov | 97009 |
| link.springer.com | 85357 |
| www.ipl.org | 84084 |
| pubmed.ncbi.nlm.nih.gov | 68934 |
| www.reference.com | 61658 |
| www.bartleby.com | 60097 |
| quizlet.com | 56752 |

Table 2: Top 10 most web domain of the data in KNOWLEDGE PILE, most of these are academic institutions, high-value forums, and authoritative website.

rums, and authoritative websites in specific knowledge fields. These resources are closely related to the knowledge domains we aim to collect, such as *en.wikipedia.org* and *www.semanticscholar.org*. Many previous research [Touvron et al., 2023a, Gao et al., 2021, Taylor et al., 2022b] have specifically collected data from these domains to enrich the knowledge of the training dataset. To gain an insight into the data distribution of KNOWLEDGE PILE, we randomly selected 100,000 examples and conducted the statistical analysis of their domain frequency. Figure 4 left shows the distribution of different frequency intervals classified by domain frequency. We observed that in the KNOWLEDGE PILEdatabase, the vast majority of web domains are recorded only once, and these domains also contain rich knowledge content. However, traditional manual data collection methods have limitations in

systematically collecting these scattered data, and *Query of CC* has shown its excellent data collection capabilities in this regard.

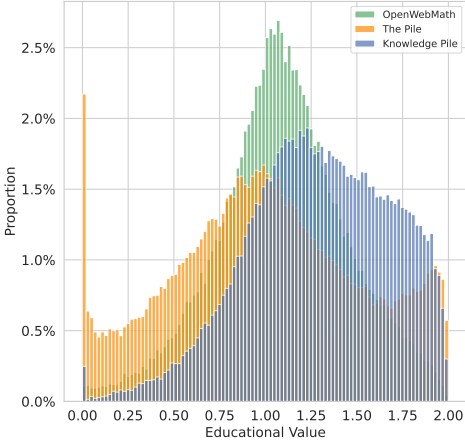

| Datasets | Educational Value (↑) |
|---|---|
| PubMed* [Cohan et al., 2018] | 1.260 |
| Guanaco* [Dettmers, 2023] | 1.115 |
| OpenWebMath* [Paster et al., 2023] | 1.089 |
| Arxiv* [Cohan et al., 2018]* | 1.068 |
| FineWeb* [Penedo et al., 2024] | 1.056 |
| Dolma v1.7* [Soldaini et al., 2024] | 1.037 |
| The Pile* [Gao et al., 2021] | 1.01 |
| MiniPile* [Kaddour, 2023] | 0.998 |
| RedPajama* [Computer, 2023] | 0.985 |
| KNOWLEDGE PILE (ours) | **1.291** |

Figure 5: The distribution of educational value of different open source datasets, which shows that the distribution of KNOWLEDGE PILE on the x-axis is significantly right shifted compare others, indicating that it has higher educational value.

Table 3: The comparison of average educational value scores among different open-source datasets. * denotes the results cited from Tsui [2024], which selected the first 100,000 samples of the dataset, and others randomly selected 100,000 samples.

Furthermore, Table 4 right statistic the timestamps of data sources in KNOWLEDGE PILE by year. It is evident that most of the data in KNOWLEDGE PILE originates from recent years, and the proportion of earlier timestamps is gradually decreasing. This phenomenon can be attributed to the exponential growth of internet data volume and the inherent timeliness characteristic of the knowledge pile.

**Data Quality Analysis** To evaluate the quality of KNOWLEDGE PILE, we employed an open-source data quality classifier[2], to rate the data within KNOWLEDGE PILE. Inspired by Gunasekar et al. [2023], high-quality data should possess characteristics of high educational value, namely: clarity, independence, instruction, and balance. To achieve the assessment of the educational value of the data, Tsui [2024] collected a subset of high-quality raw data and trained a classifier for evaluating the educational value of data based on fasttext [3]. The educational value ranges from 0, indicating low educational value, to 2, indicating high educational value. In Table 3, we present a comparison between KNOWLEDGE PILE and other mainstream knowledge datasets. The results show that KNOWLEDGE PILE has an average score of 1.29, significantly outperforming other open-source knowledge-based real datasets.

Figure 5 further reveals the differences in the distribution of educational value among the KNOWLEDGE PILE, Pile, and OpenWebMath datasets. Through comparison, we can observe a distinct rightward shift in the distribution of KNOWLEDGE PILE, indicating that KNOWLEDGE PILE contains a greater amount of data with high educational value, while the proportion of low-value data is relatively lower.

## 4.3 The Improvement in Knowledge-Related Reasoning Benchmark

Based on KNOWLEDGE PILE, we further train two models: Llama2-QoC and Mistral-QoC, based on Llama2 and Mistral.

---

[2] https://huggingface.co/kenhktsui/llm-data-textbook-quality-fasttext-classifier-v2
[3] https://fasttext.cc/

|                | | MATH | GSM8K | MMLU | AGIEval | BIG-Bench Hard |
|----------------|----|-------|-------|-------|---------|----------------|
| Code-Llama     | 7B | 3.88  | 14.4  | 40.39 | 21.47   | 42.78          |
| Baichuan2-Base | 7B | 5.72  | 23.81 | 53.95 | 34.68   | 40.32          |
| Minerva        | 8B | 14.1  | 16.2  | -     | -       | -              |
| Llemma         | 7B | 14.3  | 35.94 | 47.89 | 24.14   | 48.61          |
| Qwen 2         | 7B | 10.82 | 51.4  | 57.97 | 40.37   | 22.27          |
| Llama 2        | 7B | 3.32  | 16.68 | 46.79 | 21.37   | 38.19          |
| Llama 2-QoC    | 7B | 6.2   | 28.51 | 57.02 | 30.04   | 44.82          |
| Mistral        | 7B | 11.22 | 47.31 | 64.06 | 32.88   | 56.69          |
| Mistral-QoC    | 7B | **17.48** | **55.27** | **65.71** | **45.24** | **57.81** |

Table 4: Main results of our model and baselines in some mathematical reasoning tasks and knowledge related reasoning tasks.

**Training Details**   In our experiment, we employed the InternLM[4] [InternLM-Team, 2023] library for training all models on 256 A800 GPUs with bfloat16 mixed precision, and only utilized data parallelism during the training process. To enhance throughput and reduce memory consumption, we introduced the Flash attention 2 [Dao, 2023] module. Above these, both Llama-QoC and Mistral-QoC achieve 4000 tokens per GPU per second(TGS). More training details will be described in Appendix.

**Evaluation**   During the evaluation, we utilized the open-source library OpenCompass [5], which serves as a platform for evaluating LLMs. Leveraging Opencompass, we compare the performance with some open source pre-trained models: Llama2 [Touvron et al., 2023b], Code-Llama [Rozière et al., 2023], Baichuan 2-Base [Yang et al., 2023], Mistral [Jiang et al., 2023], Qwen 2 [Bai et al., 2023] and some language models for mathematical reasoning: Llemma [Azerbayev et al., 2023] and Minerva [Lewkowycz et al., 2022]. For the selection of evaluation datasets, we opted for three distinct capabilities to assess both Llama2-QoC and Mistral-QoC. These encompassed mathematical reasoning datasets such as Math, GSM8K, knowledge-oriented language understanding datasets including MMLU, AGIEval, and challenging reasoning tasks BIG-Bench hard. More details for evaluation will be described in Appendix.

**Main Results**   The main results of our two models trained on KNOWLEDGE PILE ( Llama2-QoC and Mistral-QoC) and the baseline are compared in Table 4 across several general benchmarks. Overall, both models exhibit significantly improved performance, particularly Mistral-QoC. In the complex mathematical reasoning benchmark MATH dataset, Mistral-QoC demonstrates a notable enhancement after QoC training, rising from 11.22 to 17.48, which surpasses professional models such as LLEMMA and Minerva by 3 points (14.3 vs 17.48). Furthermore, Mistral-QoC achieves even higher performance on the mathematical application problem GSM8K (47.31 vs 55.27). Turning to various knowledge-based reasoning tasks, Mistral-QoC displays outstanding capabilities in both MMLU and AGIEval. On the challenging BIG-Bench Hard evaluation set, the model also exhibits a noteworthy improvement in handling complex reasoning tasks.

In comparison to the backbone model, LLAMA-QoC and Mistral-QoC show substantial improvements in mathematical and knowledge-based reasoning tests. For instance, Mistral achieves a 6-points improvement in the MATH dataset and a 5-point improvement in GSM8K. In knowledge-based tests, MMLU shows a relatively modest improvement, within a 1.7 point. However, in AGIEval, Mistral-QoC outperforms Mistral by an impressive 13 points.

Otherwise, an interesting observation is that when the baseline model performance is lower (e.g., LLAMA2), the enrichment of the dataset leads to higher improvements. Conversely, for baseline models with better performance, achieving significant improvement becomes relatively challenging. This difference may be attributed to variations in the model's ability to fit the data.

---

[4]https://github.com/InternLM/InternLM
[5]https://github.com/open-compass/opencompass

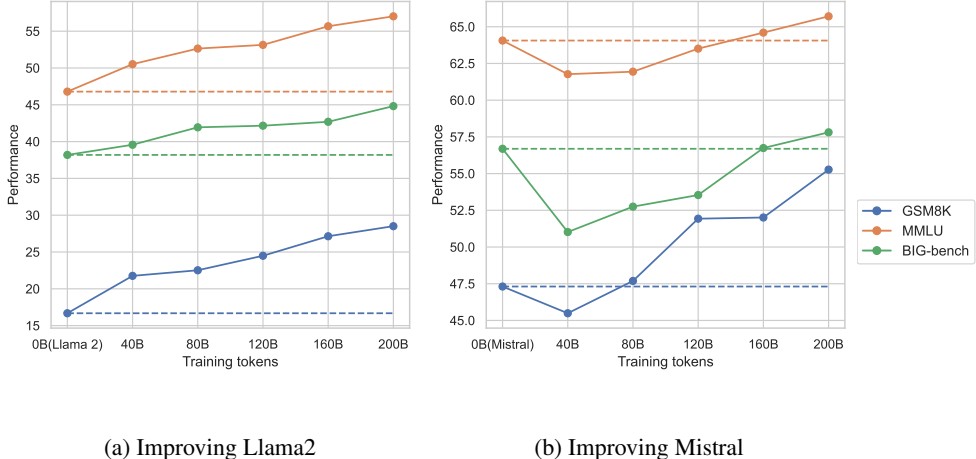



(a) Improving Llama2        (b) Improving Mistral



Figure 6: The performance curves of Llama2-QoC and Mistral-QoC, varying with the increase in the number of training tokens.

### 4.4 Difference between Improve Llama and Mistral

As shown in Table 4, it is evident that Llama2 exhibits inferior performance relative to Mistral. However, this also highlights a greater potential for performance improvement when training in KNOWLEDGE PILE. Also, we observe a significant behavioral difference during the improvement process, which is shown in Figure 6. We found that in certain tasks, Mistral's performance undergoes a certain degree of decline during the improvement process, followed by a subsequent ascent after some time, eventually surpassing the previous performance levels. This phenomenon may come from a conflict between the potential distribution of the model and the distribution of high-quality datasets. Mistral's potential distribution is superior but more unstable, whereas Llama's performance is relatively poorer but exhibits greater plasticity. On the other hand, the improvement achieved on more powerful models (Mistral 7B) also demonstrates the high-quality of KNOWLEDGE PILE.

## 5 Hallucination Analysis

The output of LLMs generally exhibits significant hallucination issues. Previous work has shown that LLMs are prone to hallucinations, and using their output directly in training without filtering can exacerbate the model's hallucinations for certain issues. However, in the data collection pipeline of *Query of CC*, the model is only used to generate queries and is not directly applied in training. Therefore, even if the synthesized queries from the LLMs contain incorrect information, the information retrieved from the corpus based on these incorrect queries is correct. Hallucinatory queries do not lead to the retrieval of incorrect information.

## 6 Conclusion

In this study, we propose an efficient method *Query of CC*, for the automated collection of specialized domain data. Leveraging seed data from some specific domains, we employ a language model for query bootstrapping. By optimizing the breadth and depth of queries, we expand the query to retrieve data relevant to the specified domain. Ultimately, we collected and released an open dataset comprising approximately 735GB of KNOWLEDGE PILE, equivalent to approximately 188 billion tokens, in the fields of mathematics and knowledge. Experimental results demonstrate that the adoption of KNOWLEDGE PILE significantly enhances the model's performance in some reasoning tasks, such as math word problems and professional examinations. Our objective is not only to establish a research foundation for community studies in mathematical and knowledge-related reasoning but also to provide an efficient and cost-effective method for collecting high-quality data, thereby facilitating the accumulation of more high-quality data.

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
