# Supplementary Material for
## *Query of CC*: Unearthing Large Scale Domain-Specific Knowledge from Public Corpora

**Zhaoye Fei[1,2♦], Yunfan Shao[1,2♦], Linyang Li[1,2♦], Zhiyuan Zeng[1,2]**
**Conghui He[2], Hang Yan[2], Dahua Lin[2] and Xipeng Qiu[1]**
[1]School of Computer Science, Fudan University, Shanghai, China
[2]Shanghai AI Laboratory
{zyfei20, yfshao19, xpqiu}@fudan.edu.cn
{lilinyang, yanhang}@pjlab.org.cn

## A Model Training Details

In this study, we train all models with bf16 mixed precision and only data parallel. We employed the AMSP [Chen et al., 2023], shard optimizer state across 8 cards to reduce communication overhead. Simultaneously, data parallelism was employed for training. To enhance throughput and reduce memory consumption, we introduced the Flash attention 2 [Dao, 2023] module. All models underwent training for 50,000 steps, with a global batch size of 4 million tokens per step, totaling 200 billion training tokens.

During the initial 2,000 steps of training, the learning rate was warmed up to the maximum, and then, at the end of training, it was decayed according to cosine decay to the specified minimum learning rate. Specifically, for Llama2-QoC, the maximum learning rate during training was set to 2e-5, and the minimum learning rate was set to 2e-6. For Mistral-QoC, the maximum learning rate during training was 5e-6, descending to 2e-7 at the end of training. In our experiment, Llama2-QoC and Mistral-QoC achieved a training throughput of approximately 4000 tokens per GPU per second(TGS). Despite the relatively higher training efficiency of Llama2-QoC, it still utilized 14,000 GPU hours.

## B Evaluation

During the evaluation, we utilized the open-source library OpenCompass [2], which serves as a platform for evaluating large models. OpenCompass offers various evaluation datasets and supports efficient task partitioning to maximize the utilization of computational resources. For the selection of evaluation datasets, we opted for three distinct capabilities to assess both Llama2-QoC and Mistral-QoC. These encompassed mathematical reasoning datasets such as Math, GSM8K, knowledge-oriented language understanding datasets including MMLU, AGIEval, and challenging reasoning tasks BIG-Bench hard. The details of evaluation datasets are as follows:

**Math [Hendrycks et al., 2021a]** Math datasets comprising 12,500 competitive mathematical problems spanning challenging areas such as algebra and number theory. During evaluation, we selected four questions as examples, and each illustrating complete steps of problem-solving approaches. We assessed the model-generated answers for equivalence with these golden answers.

**GSM8k [Cobbe et al., 2021]** GSM8k datasets contain 8.5k high-quality grade school math word problems, the task requires the large language model answered to combine world knowledge and mathematical reasoning. In this task, following xxx, we provide four questions and the solution with more detail as examples and also evaluate the equivalence of generated answer with the golden answer.

---

[2]https://github.com/open-compass/opencompass

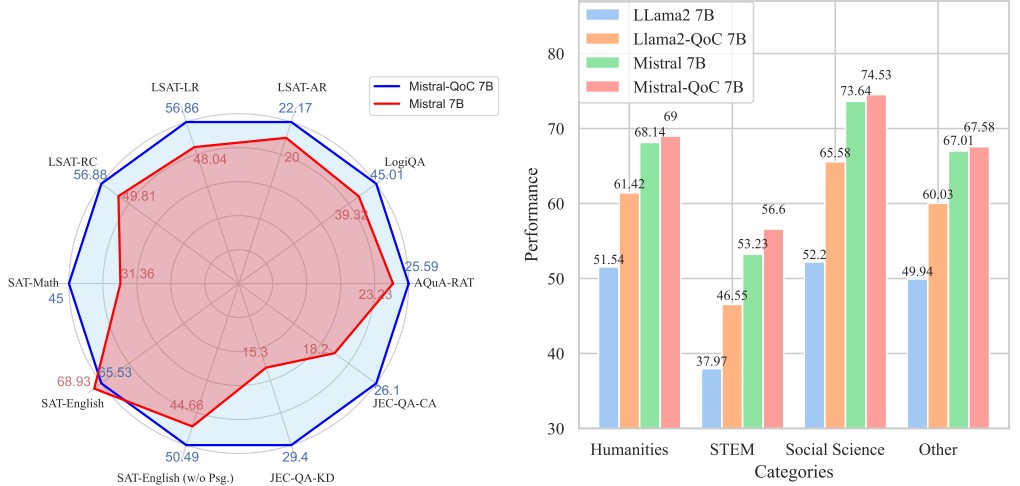

Figure 1: Left: the comparison of performance in AGIEval Benchmark between Mistral and Mistral-QoC. Right: The performance comparison of performance in MMLU.

**MMLU [Hendrycks et al., 2021b]**   MMLU is a vast multi-task dataset encompassing questions from various disciplines, including humanities, social sciences, STEM, and others. There are 57 sub-tasks in MMLU including elementary mathematics, American history, computer science, law, etc. During the evaluation, we provided five example questions and their answers with relatively complete chains of thought, using this to judge whether the model could correctly select options. This task necessitates a broad range of world knowledge and problem-solving capabilities for large language models.

**AGIEval [Zhong et al., 2023]**   AGIEval is also a benchmark that has various categories, and is designed for accessing the performance of large language models in context with human-centric standardized exams. Compared to MMLU, this dataset has more comprehensive data sources and provides a better evaluation of cross-linguistic knowledge performance.

**BIG-Bench hard [Suzgun et al., 2023]**   BIG-Bench hard is a challenging subset of BIG-Bench, which is designed to evaluate the reasoning ability of large language models. This dataset comprises 23 BIG-Bench tasks covering different programming languages and domains. During the evaluation, we provide three examples of large language models and assess whether the model could accurately answer the presented problems.

## C   More Evaluation of Llama-QoC and Mistral-QoC

In order to compare the effects of KNOWLEDGE PILE on knowledge-related reasoning abilities, Figure **??** contrasts the improvement based on Llama2 and Mistral on MMLU, with a specific focus on performance across different subjects.

From the perspective of Llama-QoC and Llama, following training with KNOWLEDGE PILE, the average performance on MMLU for all categories increased by at least 10 points. This notable performance enhancement can be attributed in part to the initially lower performance of Llama, while the KNOWLEDGE PILE exhibits relatively rich knowledge content, playing a significant role in driving the improvement of Llama2's performance.

In comparison to Llama, Mistral's performance improvement is relatively moderate. The results indicate that in the STEM field, Mistral-QoC demonstrates a higher performance improvement compared to other fields, rising from 53.23 to 56.6. In contrast, performance improvements in other categories hover around one point. Two possible reasons account for this phenomenon. Firstly, Mistral exhibits strong comprehension abilities in disciplines such as humanities and social sciences, consistently scoring above 65, leaving relatively limited room for improvement. In contrast, the model's score in the STEM field is only 53.23, suggesting a greater potential for performance

enhancement. Secondly, KNOWLEDGE PILE has the largest and most extensive dataset in the STEM field, contributing to the more pronounced performance improvement of Mistral-QoC on MMLU-STEM.

Figure **??** compares the Performance of professional and academic exams covered in AGIEval. Even in the case of professional academic exams, KNOWLEDGE PILE similarly leads to substantial performance improvements. As observed in the figure, apart from a slight decline in the SAT-English expression test (from Mistral 68.93 to 65.53), other tasks exhibit noticeable improvements after continued training in KNOWLEDGE PILE. Notably, SAT-Math rises from 31.36 to 45, which is a significant improvement for Mistral. Legal exams such as JEC-QA-CA and JEC-QA-KD also show significant enhancements, with the performance increasing from 18.2 and 15.3 to 26.1 and 15.3. For other law-related exam, LSAT is divided into three aspects: Law-Analytics (LSAT-AR), Law-Logic(LSAT-LR), and Law-Reading(LSAT-RC) improvements of 2.17, 8.82, and 7.07 point, respectively. For the LogiQA, Mistral-QoC achieves an accuracy of approximately 45.01%, nearly a 20% improvement compared to Mistral (39.32)

## D    Collection implement details

In the implement section, we primarily discuss three components, the public corpora (D.1), retrieval engine (D.2), and the post-processing settings (D.3).

### D.1    Public Corpora

With the development of large language models, public corpora have become increasingly rich, including the Pile, RedPajama, and Common Crawl. Common Crawl is an open-source web crawler project containing all publicly available web pages from 2013 to the present. In theory, it encompasses a significant portion of the information present on the web. Due to hardware cost constraints, we utilized WARC format data from the years 2016 to 2023 to build our retrieval database. We performed extraction, filtering, and cleaning procedures on the data obtained from Common Crawl to ensure the quality of the retrieval dataset. The final retrieval database comprises several billion records, occupying a total of 50TB of disk space.

### D.2    Retrieval Engine

Retrieving data from billions of documents is highly challenging, and we need to calculate the relevant score between queries and each document in the retrieved database. To improve storage and retrieval efficiency, we built the retrieval engine based on Elasticsearch (ES). Elasticsearch is an open-source distributed search engine that employs distributed storage and inverted indexes, achieving data retrieval highly efficient. We selected the BM25 algorithmic as the relevance scorer because of its efficiency. For each query, we recall the top 1000 most relevant documents. Leveraging Elasticsearch's efficient storage and indexing algorithms, we can complete a query retrieval within 100ms.

### D.3    Post Processing

To enhance the quality of KNOWLEDGE PILE, inspired by Wenzek et al. [2020], we conducted data quality filtering and deduplication in the post-processing stage.

In the data quality filtering phase, we manually selected some high-quality data as positive examples and randomly selected low-quality data from Common Crawl, such as poorly structured data and advertising data, as negative examples to train our scoring model. High-quality data mainly includes papers, books, and high-quality forum data. We chose BERT-base-uncased as the backbone to train the model and tested it on a subset of data to ensure its high usability. Finally, we scored the retrieved data and filtered out data with quality scores below 0.8.

In the deduplication phase, we employed the Minhash-LSH method. For hyperparameter selection, we computed the similarity scores based on 13 grams and set the similarity threshold to 0.8. Additionally, we set $num\_perm$ to 128 to balance computation efficiency and deduplication performance.

# E  Limitation

In this paper, we propose a collection method for domain-specific data, *Query of CC*, which extends domain-relevant queries through LLM bootstrapping for data retrieval. Additionally, we demonstrate through empirical evaluation on further training of LLAMA and Mistral that data collected using this method significantly improves the ability of LLM in some specific domains. However, this method still has the following potential limitations:

**Data Quality**  The quality of data collected by *Query of CC* depends largely on the quality of data in public corpora. In this work, we utilized the Common Crawl corpus as our public corpus, extracting and processing the CC dump up to April 2023. Despite leveraging some methods [Wenzek et al., 2020] for processing high-quality web data, we found that the corpus still contains an amount of low-quality and erroneously extracted content. Thus, improving the data quality of public corpora is also a direction for future research.

**Data Scale**  The data source of *Query of CC* is limited by public corpora, and unable to collect information not present in these corpora. This limitation requires particular attention. Moreover, the proportion of domain data in public databases also influences the scale of the collected dataset. During the collection process, scoring based on similarity scores can be employed to filter highly relevant data.

# F  Prompt of Query Bootstrapping

================= PROMPT OF QUESTION EXTENSION =================

Suppose you are a question creator and your task is to create a new question based on the example question! Note that the new questions should have the same domain as the example questions, but be less frequent, of exactly the same length and difficulty as the example questions. You need to use your ingenuity to create a problem that is completely different from the given problem.
Sample questions will be given after ###Given Question###. You need to write the newly created question after ###Created Question###.
###Given Question###

[question]

================= PROMPT OF THOUGHT GENERATION =================

Suppose you are a expert and your task is answer the given problem and tell me how to get the answer! You need to write the answer after the ###Answer### symbol. Please write the chain of thought after the ###COT### symbol.
###Given Question###

[question]