# OpenReview forum: "Query of CC: Unearthing Large Scale Domain-Specific Knowledge from Public Corpora"
_NeurIPS.cc/2024/Datasets_and_Benchmarks_Track — Submitted to NeurIPS 2024 Track Datasets and Benchmarks_

### Official Review · Reviewer_uPAi · 2024-07-24
**Review of Query of CC: Unearthing Large Scale Domain-Specific Knowledge from Public Corpora**

**Rating:** 7
**Confidence:** 2
**Correctness:** The methods appear generally sound.
**Clarity:** The paper is well written and well or…

**Review:**

The project presents a new method of automated data collection (from the Common Crawl corpus) to retrieve data from specific information domains that can then be applied to enhance general LLMs. Its success offers a promising potential platform for future researchers to leverage and an interesting new departure point for future work on automated, domain-specific data collection.

**Strengths:**

The paper is clearly written and the project well structured. The method is promising and a likely platform for further work across a vast range of disciplines.

**Additional Feedback:**

No additional feedback.

**Documentation:**

Methods and data are presented in sufficient detail to facilitate reproducability and use by other researchers.

**Ethics:**

I have no ethics concerns with this project.

**Limitations:**

The paper would benefit from more discussion of limitations of the method, particularly likely differences in usefulness across domains.

**Opportunities For Improvement:**

The authors might consider adding (speculation) regarding those LLM tasks where automated collection rather than manual is most likely to prove beneficial and vice versa.

**Relation To Prior Work:**

The paper clearly situates the project within the relevant literature and past work.

**Summary And Contributions:**

The project presents a new method of automated data collection (from the Common Crawl corpus) to retrieve data from specific information domains that can then be applied to enhance general LLMs. The ability to automate retrieval of domain-specific information saves research time spent on data collection, apparently without significant sacrifice in utility of corpora collected.

---

> ### Author Rebuttal · Authors · 2024-08-16
>
> Thank you for your positive feedback and professional insights on our work. We appreciate your thoughtful suggestions, which offer valuable direction for future research. As you mentioned, the effectiveness of automated versus manual data collection in various tasks is certainly an area worth exploring further. We agree that this is an important topic, and we plan to investigate it more deeply in our future work.
>
> If you have any other questions，lease feel free to comment,  and we would be pleasure to respond as quickly as possible.

---

### Official Review · Reviewer_E2JB · 2024-07-25

**Rating:** 6
**Confidence:** 5
**Correctness:** The dataset lacks proper baselines.
**Clarity:** Yes.

**Review:**

### Pros:

1. This paper starts with keywords to gradually augment queries in an automatic manner, and then retrieve relevant documents with BM25. The proposed approach is simple, and differs from previous works that typically require some seed datasets to retrieve.
2. The resulting dataset Knowledge Pile is already publicly available, that could benefit the community.

### Cons:

As a scientific study, I don’t think the evidence is enough to support the proposed method QueryCC is a superior approach to collect continual training corpus – the comparison in the experiments lacks proper baselines. For example, the random baseline should be reported, and there could be some simple baselines as well like using general SFT datasets to retrieve documents that is even simpler than the proposed approach.

**Strengths:**

In addition to the pros listed above, the released dataset could be a good resource to the community.

**Additional Feedback:**

NA

**Documentation:**

Yes.

**Ethics:**

No.

**Limitations:**

The paper didn’t discuss limitations.

**Opportunities For Improvement:**

1. As mentioned in the cons above, I think proper baselines should be added to support the proposed approach.
2. In terms of writing, in Section 4.2 Data Analysis section, I suggest clearly listing the dataset size in the table such as the number of tokens – this important information is only found in the experiment's figures and the conclusion section if I didn’t miss something.

**Relation To Prior Work:**

Yes;

**Summary And Contributions:**

This paper proposes a method, query of CC, to start with a few keywords and bootstrap to generate queries based on these keywords. Then the queries are used to retrieve relevant documents with BM25. The retrieved corpus, named as Knowledge Pile, is adopted as the corpus to continually pretrain mistral and llama 2 models. Experiments on several knowledge and reasoning benchmarks demonstrate effectiveness of the proposed method.

---

> ### Author Rebuttal · Authors · 2024-08-16
>
> Thank you for your valuable suggestions regarding the baselines in our work. As mentioned by Reviewer 4CSd, training the baselines is indeed a resource-intensive task. In our paper, we stated that training the Llama-QoC (7B) model required approximately 14,400 A800 GPU hours. Otherwise, some research indicates that the performance improvements of large language models often plateau during the later stages of pre-training. However, our data demonstrated a significant performance boost during the post-training phase, which, to some extent, underscores the high quality of our dataset.
>
> Otherwise, we conducted a smaller-scale experiment comparing the performance of models trained from scratch using the 2B LLaMA architecture on 200B tokens from the Knowledge Pile and C4.
>
> | Dataset        |  MMLU |
> |----------------|:-----:|
> |C4    | 28.52 |
> | KNOWLEDGE PILE | 33.13 |
>
> The results, show that during the pre-training phase, the Knowledge Pile outperformed other datasets based on random samples from Common Crawl (CC) in terms of performance.
>
> Moreover, while using a general SFT dataset for retrieval is a simple and effective approach, these general datasets are often constrained by their specific domains. In contrast, the Query of CC (QoC) does not suffer from such limitations, making it a more versatile option for domain-specific retrieval.
>
> For your advice in `Opportunities For Improvement`, we will enhance the details regarding the dataset tokens in Table 1 of the revised manuscript and adding some baseline.
>
> Finally, I hope this response addresses your concerns. Should you have any further questions or require additional clarification, please feel free to comment, and we will respond as promptly as possible.

---

> > ### Comment · Reviewer_E2JB · 2024-08-29
> >
> > Thanks for your reply. While I understand the expensive resources to run baselines, indicating proper baselines even in a smaller scale is very necessary, otherwise someone can just create new datasets with heuristics and how can we judge the merit and contributions of such datasets? I appreciate the added comparison of the 2B model on MMLU, in future revisions, I encourage the authors to include more benchmarks on the comparison. Based on the added results, I would like to increase my score to 6.
> >
> > Even on a smaller scale, for example, I think it is quite feasible and reasonable to train 1B-size models on 50B tokens to show the strengths of the proposed dataset.

---

### Official Review · Reviewer_4CSd · 2024-08-05
**Interesting method for Pre-Training Corpus generation with some problems in presentation and evaluation.**

**Rating:** 6
**Confidence:** 4

**Review:**

See other points.

**Strengths:**

-	A large-scale knowledge dataset for training LLMs is proposed. This is a highly relevant topic and could be interesting for further research.
-	The methodology for curating the dataset is innovative and is an interesting new direction for automatic curating of pre-training corpora. The idea itself is simple and scalable.
-	The paper is well-written and easy to follow.
-	The experiments are showing good performance of models trained on the proposed dataset. However, I have some doubts about possible data contamination.

**Additional Feedback:**

No additional feedback

**Clarity:**

-	It is unclear if only English data was used.
-	The description of the methodology for creating the dataset is very vague. The explanation on how the seed queries were collected is completely missing. The details of the bootstrapping are unclear and could be supported by some examples.
-	What LLM was used for the question generation/thought generation?
-	‘In our experiments, the typical choice for k is 1000’ – What do you mean with typical choice?
-	“other miscellaneous.” Are these actual keywords or is the list just longer and you did not put the whole list in there.
-	Are the documents somehow chunked or is each website a single document in the retrieval process?
-	How do you determine how big the dataset should be? The number of generated queries and a choice of k=1000 seems to be arbitrary.

**Correctness:**

-	Data contamination by benchmark data is not analyzed nor is the data filtered accordingly to prevent contamination. This could be the reason for the significant increase in performance on the benchmark datasets and this should be controlled/analyzed somehow. Ideas for further analysis are described in [1].

[1] Jiang, M., Liu, K. Z., Zhong, M., Schaeffer, R., Ouyang, S., Han, J., & Koyejo, S. (2024). Investigating data contamination for pre-training language models. arXiv preprint arXiv:2401.06059.

**Documentation:**

The dataset is available, however, there is not enough details for reproducibility.
The code that was used for the dataset generation is not public.

**Ethics:**

Issues about data privacy, copyright, consent, data quality and representativeness, as well as discrimination, bias and fairness, are not discussed.

**Limitations:**

-	No discussion on ethical considerations and no dataset datasheet. I found the dataset datasheet on the GitHub repo which is not linked in the paper.
-	No educational value of the other open-source datasets from Table 1 are provided. Why not?
-  The potential societal impact of this work is not discussed, but should be. This is a large new pre training corpus for LLMs and it may include lots of biased, discriminating and false data. This should be discussed.

**Opportunities For Improvement:**

-	Differences to manually curated datasets are unclear
- The performance of the two trained models is not compared to training on other public datasets. However, training a baseline model on different datasets is unfeasible due to the amount of GPU hours required to train such a model.
-	Section 5 seems kind of unnecessary. It would be more interesting to see how much the query generation/bootstrapping actually improves the quality of the dataset. Would the models also get better when just trained on a random subset of CC? Would a simple BM25 baseline with keywords work similarly well? The effects of the approach are very hard to estimate when no ablation study is performed.
- More details need to be mentioned to make the work reproducible.
- More analysis on the quality, diversity, and inclusion of private information in the dataset should be performed.
- The current process does not contain any post-processing steps for quality assurance. Or does it? The supplements mention a BERT classifier for quality assurance which is never mentioned in the main paper.

**Relation To Prior Work:**

-	Imitation learning models that also use automatic training data generation methods to enhance reasoning capabilities of models are related and should be discussed. [1]
-	Differences to other public training corpora should be discussed.

[1] Mukherjee, S., Mitra, A., Jawahar, G., Agarwal, S., Palangi, H., & Awadallah, A. (2023). Orca: Progressive learning from complex explanation traces of gpt-4. arXiv preprint arXiv:2306.02707

**Summary And Contributions:**

The paper presents a methodology based on LLM-generated queries to create a domain-specific pre-training corpus for LLMs.
The method starts with a set of seed queries, which are used to create questions with an LLM and then used to create “thoughts”. These are used for document retrieval with BM25. The retrieved documents are then used as the dataset for training of two open LLMs and compared to other open LLMs on standard benchmarks.
They improve the performance of other LLMs significantly.

---

> ### Author Rebuttal · Authors · 2024-08-16
>
> We sincerely thank you for your detailed and constructive feedback. Your comments are of great significance to me in improving the quality of my paper. Next, I will respond to your comments and questions one by one.
>
> Opportunities For Improvement:
>
> > Q1 Differences to manually curated datasets are unclear
>
> We apologize for the lack of clarity in the manuscript regarding this aspect. The primary difference between Query of CC and manually curated datasets lies in the reduction of human effort required for collecting domain-specific data. Manually curated domain-specific datasets typically require significant human labor for collection and filtering. For instance, the Pile dataset comprises 22 different web domains, each of which necessitates extensive human effort for collection, formatting, and initialization. The OpenWebText dataset employs several filters designed to extract high-quality, domain-specific texts. In contrast, Query of CC allows for the collection of high-quality, domain-specific data by merely providing relevant keywords, thus minimizing the need for manual intervention.
>
> > Q2 The performance of the two trained models is not compared to training on other public datasets. However, training a baseline model on different datasets is unfeasible due to the amount of GPU hours required to train such a model.
>
> As you correctly noted, training a baseline model is indeed costly. Recent research suggests that the performance gains of large language models tend to plateau during the later stages of pre-training, indicating diminishing returns when continuing to train on random subsets. To support this point, we conducted a smaller-scale experiment comparing the performance of models trained from scratch using the 2B LLaMA architecture on 200B tokens from the Knowledge Pile and C4 datasets.
>
> | Dataset        |  MMLU |
> |----------------|:-----:|
> |C4    | 28.52 |
> | KNOWLEDGE PILE | 33.13 |
>
> It is evident that during the pre-training phase, the Knowledge Pile outperforms other datasets obtained from random sampling of CC-based data.
>
> > Q3 Section 5 seems kind of unnecessary. It would be more interesting to see how much the query generation/bootstrapping actually improves the quality of the dataset. Would the models also get better when just trained on a random subset of CC? Would a simple BM25 baseline with keywords work similarly well? The effects of the approach are very hard to estimate when no ablation study is performed.
>
> Your concerns about the relevance and necessity of Section 5, along with the suggestion for an ablation study, highlight critical aspects that need further elaboration in our research. Regarding the improvement in dataset quality, we have reported in Q2 an experiment comparing the performance of the dataset collected via query generation/bootstrapping (i.e., “Knowledge Pile”) against datasets collected from random subsets of Common Crawl (CC) in downstream tasks. The results indicate that the “Knowledge Pile,” formed through purposeful query generation, provides more relevant and valuable data for specific tasks compared to random subsets of CC.
>
> As for your suggestion to compare with a simple BM25 baseline using keywords, we believe this is a valuable perspective that would help further validate the effectiveness of our method. In fact, the number of keywords used in our experiments was relatively small, limiting their efficacy in retrieving a large number of data, which restricted their role in the comparative analysis.
>
> > Q4 More details need to be mentioned to make the work reproducible.
>
> Thank you for your valuable feedback on the reproducibility of our work. I fully agree that detailed information is crucial for other researchers to replicate and validate our findings. To this end, we have provided a detailed description of the preprocessing steps, including the cleaning process of the public corpus, the retrieval configuration, and the post-processing steps, in the appendix of the paper. We hope these details will assist other researchers in better understanding and replicating our study. Should you have any further questions or require additional information on the reproducibility aspects, please do not hesitate to contact us. We are happy to discuss and provide support. Once again, thank you for your feedback.

---

> > ### Author Rebuttal · Authors · 2024-08-16
> >
> > > Q4 More details need to be mentioned to make the work reproducible.
> >
> > Thank you for your valuable feedback on the reproducibility of our work. I fully agree that detailed information is crucial for other researchers to replicate and validate our findings. To this end, we have provided a detailed description of the preprocessing steps, including the cleaning process of the public corpus, the retrieval configuration, and the post-processing steps, in the appendix of the paper. We hope these details will assist other researchers in better understanding and replicating our study. Should you have any further questions or require additional information on the reproducibility aspects, please do not hesitate to contact us. We are happy to discuss and provide support. Once again, thank you for your feedback.
> >
> > > Q5 More analysis on the quality, diversity, and inclusion of private information in the dataset should be performed.
> >
> > Thank you for your insightful suggestion. I understand that analyzing the quality, diversity, and inclusion of private information in the dataset is crucial for ensuring the rigor and validity of our research. These factors directly impact the reliability of our findings and the ethical use of data in our study.
> >
> > In our paper, we conducted a classifier-based quantitative analysis of data quality, providing a detailed comparison of our dataset's quality distribution relative to other datasets. For diversity, we examined the distribution of data across various web domains and temporal patterns. Regarding privacy, we have relied on the safeguards implemented during the construction of public datasets to protect private information, although this aspect warrants further scrutiny.
> >
> > In response to your suggestion, we will expand and strengthen our analysis in the following areas:
> >
> > 1. Quality Assessments: We will conduct additional evaluations using perplexity-based quality scoring[1] or QuRating[2] to further verify the data quality of the Knowledge Pile.
> >
> > 2. Diversity Analysis: We will analyze the distribution within the dataset to provide a more comprehensive assessment of its diversity.
> >
> > 3. Privacy Considerations: We will perform a detailed analysis of private information within the dataset to identify and mitigate any potential inclusion of sensitive information.
> >
> > We appreciate your constructive feedback and believe these enhancements will significantly improve the rigor of our study. We will incorporate these analyses in our revised manuscript.
> >
> > > Q6 The current process does not contain any post-processing steps for quality assurance. Or does it? The supplements mention a BERT classifier for quality assurance which is never mentioned in the main paper.
> >
> > Regarding your concern about post-processing steps for quality assurance, we would like to clarify that the main paper primarily focuses on the core methodology and experimental results. However, the BERT classifier mentioned in the supplementary materials was indeed part of our post-processing process. In our study, we introduced a BERT classifier during the post-processing phase to evaluate the quality of the generated results. However, as the classifier did not significantly filter out any low-quality texts (i.e., all texts were deemed acceptable), we decided to omit this section from the main text to streamline the discussion and focus on the core findings.
> >
> > [1] Alexander Wettig, Aatmik Gupta, Saumya Malik, and Danqi Chen. Qurating: Selecting high-quality data for training language models. CoRR, abs/2402.09739, 2024. doi: 10.48550/ARXIV.2402.
> > 9739. URL https://doi.org/10.48550/arXiv.2402.09739.
> >
> > [2] Guillaume Wenzek, Marie-Anne Lachaux and Alexis Conneau etc. Ccnet: Extracting high quality monolingual datasets from web crawl data. Proceedings of The 12th Language Resources and Evaluation Conference, LREC 2020, Marseille, France, May 11-16, 2020, pages 4003–4012. European Language Resources Association, 2020. URL https://aclanthology.org/2020.lrec-1.494/.

---

> > > ### Author Rebuttal · Authors · 2024-08-16
> > >
> > > Limitations:
> > >
> > > > Q1 No discussion on ethical considerations and no dataset datasheet. I found the dataset datasheet on the GitHub repo which is not linked in the paper.
> > >
> > > Our work primarily focuses on the collection of high-quality data in specific domains. However, we recognize the importance of ethical considerations, as you mentioned. We will address this in the revised manuscript and include the dataset datasheet in the appendix.
> > >
> > > > Q2 No educational value of the other open-source datasets from Table 1 are provided. Why not?
> > > We will include a comparison of the educational value of the open-source datasets listed in Table 1 in the subsequent revisions.
> > >
> > > | Dataset        | Educational Value (↑) |
> > > |----------------|-----------------------|
> > > |PILE          | 1.011                 |
> > > | OpenWebMath    | 1.089                 |
> > > | Proof of Pile  | 1.13                  |
> > > | MATHPILE       | 1.25                  |
> > > | KNOWLEDGE PILE | 1.29                  |
> > >
> > > > Q3 The potential societal impact of this work is not discussed, but should be. This is a large new pre-training corpus for LLMs and it may include lots of biased, discriminating and false data. This should be discussed.
> > >
> > > Thank you for highlighting the societal impact of our work, particularly the potential inclusion of biased, discriminatory, or false data in a large pre-training corpus for LLMs. We acknowledge the importance of addressing these issues and will include a dedicated section in the revised manuscript to discuss them. This section will outline the potential risks associated with large-scale data collection, including the amplification of biases present in the data. We will also discuss strategies to mitigate these risks, such as filtering, balancing datasets, and incorporating fairness-focused evaluation metrics. Furthermore, we will emphasize the importance of ongoing research in this area to reduce the impact of biases and misinformation in LLMs. We believe that by addressing these issues, we will not only strengthen our paper but also contribute positively to broader discussion on responsible AI development.
> > >
> > > Correctness:
> > > > Q1 Data contamination by benchmark data is not analyzed nor is the data filtered accordingly to prevent contamination. This could be the reason for the significant increase in performance on the benchmark datasets and this should be controlled/analyzed somehow. Ideas for further analysis are described in [1].
> > > >
> > > >[1] Jiang, M., Liu, K. Z., Zhong, M., Schaeffer, R., Ouyang, S., Han, J., & Koyejo, S. (2024). Investigating data contamination for pre-training language models. arXiv preprint arXiv:2401.06059.
> > >
> > > Thank you for your attention and feedback on our work. We acknowledge that data contamination is an issue worth considering, and we will delve into this further in future research. We will introduce the analysis you mentioned to address data contamination in the subsequent versions.

---

> > > > ### Author Rebuttal · Authors · 2024-08-16
> > > >
> > > > Clarity:
> > > >
> > > > Thank you very much for your suggestions on improving the clarity of our paper. These will certainly make our manuscript clearer and more user-friendly. We will address the following issues in the revised version:
> > > >
> > > > > Q1 It is unclear if only English data was used.
> > > >
> > > > Yes, we only included English data, and we will add this in our revision paper.
> > > >
> > > > > Q2 The description of the methodology for creating the dataset is very vague. The explanation on how the seed queries were collected is completely missing. The details of the bootstrapping are unclear and could be supported by some examples.
> > > >
> > > > We will include examples to better illustrate the process.
> > > >
> > > > > Q3 What LLM was used for the question generation/thought generation?
> > > >
> > > > We used the LLaMA2 13B chat as our LLM for question and thought generation.
> > > >
> > > > > Q4 ‘In our experiments, the typical choice for k is 1000’ – What do you mean with typical choice?
> > > >
> > > > We apologize for not making this clear in the manuscript. The choice of k=1000 is derived from a common configuration in information retrieval[1], where 1000 candidates are often used as an empirical setting in many retrieval systems. We believe that the goal of retrieving high-quality data aligns with the objectives of initial retrieval, which is to ensure the inclusion of a sufficient amount of relevant content. Therefore, we chose 1000 as document count of single query retrieved. We apologize once again for not clearly describing this choice.
> > > >
> > > > > Q5 “other miscellaneous.” Are these actual keywords or is the list just longer and you did not put the whole list in there.
> > > >
> > > > Thank you for pointing out this mistake. The “miscellaneous” section indeed contains only those specific keywords：medicine, virology and commonsense knowledge .
> > > >
> > > > > Q6 Are the documents somehow chunked or is each website a single document in the retrieval process?
> > > >
> > > > We apologize for the confusion. In the dataset constructed from Common Crawl, each webpage is treated as a single document, with no chunking performed.
> > > >
> > > > > Q7 How do you determine how big the dataset should be? The number of generated queries and a choice of k=1000 seems to be arbitrary.
> > > >
> > > > In our dataset construction process, the size of the dataset is primarily determined by the number of generated queries, with the number of documents retrieved per query fixed at 1000. It is important to note that the retrieved data will contain some duplicates, resulting in a final dataset size smaller than number of queries * k. The value of k in our experiment is not arbitrary but is a common configuration used in retrieval tasks. We provide a detailed explanation of this choice in Clarity-Q4.
> > > >
> > > > [1] Tri Nguyen, Mir Rosenberg and Xia Song etc. MS MARCO: A human generated machine reading comprehension dataset. Proceed-ings of the Workshop on Cognitive Computation: Integrating neural and symbolic approaches 2016 co-located with the 30th Annual Conference on Neural Information Processing Systems (NIPS 2016), Barcelona, Spain, December 9, 2016, volume 1773 of CEUR Workshop Proceedings. CEUR-WS.org, 2016. URL https://ceur-ws.org/Vol-1773/CoCoNIPS_2016_paper9.pdf.7
> > > >
> > > >
> > > > Finally, thank you once again for your thorough review of our work. If you have any further questions, please feel free to comment, and we will be happy to address them as soon as possible.

---

> > > > > ### Comment · Reviewer_4CSd · 2024-08-29
> > > > > **Updated Scores**
> > > > >
> > > > > I thank the authors for the extensive and helpful rebuttal.
> > > > > I appreciate that you provided new baseline results. I think these are a good showcase that the technique actually provides a better quality pre-training dataset.
> > > > >
> > > > > I would be happy also to see an analysis of data contamination.

---

### Official Review · Reviewer_menD · 2024-08-08
**Interesting idea, very poor execution**

**Rating:** 4
**Confidence:** 4

**Review:**

Overall, the procedure described in the paper looks sound and useful, provided that it is novel enough.
However, clarity, correctness, and lack of documentation are big concerns (see below).

**Strengths:**

The methodology of using LLMs to automatically generate queries sounds promising.

**Additional Feedback:**

I do not have additional feedback to give.

**Clarity:**

The paper is not clear.

1. The manuscript is compiled with the `preprint` option, which removes line numbers from the output file.
1. A great number of references are to preprints available only on ArXiv.
1. The references include links to GitHub and Hugging Face, which should instead clearly marked as footnotes.
1. There are multiple instances of words used incorrectly, inconsistent capitalization, article usage, and incorrect punctuation.
1. There are non-standard and unclear expressions that are not explained. For instance, "data recipe", "highly scattered domain data distribution", "queries around the seed keywords in both depth and breadth", "employing LLMs to generate the cognitive processes necessary for answering questions".
1. Text styles are often used in unconventional ways. The title contains words in italics, bold text is used for emphasis (instead of italics), and "Knowledge Pile" is (almost) always in teletype (`\texttt`). This makes the paper more difficult to read than it should be.
1. Figure 4 (left): this figure, with indices of an array on the x-axis, is unnecessarily hard to parse; consider a single plot with logarithmic scale y-axis.
1. Figure 4 (right): why are the bars of different colors if there is no hue?

**Correctness:**

**Dataset quality**

The main evaluation of the dataset quality ("Educational Value", Section 4.2) relies on methods discussed in not peer-reviewed publications.
Namely, these are:
```
- Ken Tsui. llm-data-textbook-quality-fasttext-classifier-v2, 2024. URL https://huggingface.co/
kenhktsui/llm-data-textbook-quality-fasttext-classifier-v2.
- Suriya Gunasekar, Yi Zhang, Jyoti Aneja, Caio César Teodoro Mendes, Allie Del Giorno, Sivakanth
Gopi, Mojan Javaheripi, Piero Kauffmann, Gustavo de Rosa, Olli Saarikivi, Adil Salim, Shital
Shah, Harkirat Singh Behl, Xin Wang, Sébastien Bubeck, Ronen Eldan, Adam Tauman Kalai,
Yin Tat Lee, and Yuanzhi Li. Textbooks are all you need. CoRR, abs/2306.11644, 2023. doi:
10.48550/ARXIV.2306.11644. URL https://doi.org/10.48550/arXiv.2306.11644.
```
Moreover, the numbers reported in Table 3 also come from [Tsui]. The authors do not report checking these numbers or independently reproducing the experiments.

**Model evaluation**

The evaluation of the models, included to show the superiority of the proposed dataset, lacks credible baselines.
Indeed, the paper compares models trained on the proposed  domain-specific dataset and base models.
A comparison with models trained on other datasets (such as the ones in Section 2) is in my opinion necessary.

**Documentation:**

The dataset is not documented properly.
Extract from the NeurIPS 2024 call for datasets:
```
Submission introducing new datasets must include the following in the supplementary materials (as a separate PDF):
- Dataset documentation and intended uses. Recommended documentation frameworks include datasheets for datasets, dataset nutrition labels, data statements for NLP, data cards, and accountability frameworks.
[...]
- Author statement that they bear all responsibility in case of violation of rights, etc., and confirmation of the data license.
- Hosting, licensing, and maintenance plan. The choice of hosting platform is yours, as long as you ensure access to the data (possibly through a curated interface) and will provide the necessary maintenance.
```

The supplementary materials do not include the dataset documentation, the intended uses, the author statement of responsibility, nor the hosting, licensing, and maintenance plan.

Additionally, the dataset is currently hosted on Hugging Face and requires the user's email to access it, potentially breaching reviewer anonymity.

**Ethics:**

No.

**Limitations:**

They are described in Appendix E, but they should be in the main text.

**Opportunities For Improvement:**

- Adhere to the submission format and guidelines.
- Improve English, at least in the choice of words and punctuation.

**Relation To Prior Work:**

It is discussed.

**Summary And Contributions:**

The authors propose a new method to retrieve data from a public corpus and use it to build a new domain-specific dataset.

---

> ### Author Rebuttal · Authors · 2024-08-16
>
> We sincerely appreciate the your recognition of the novelty and significance of our research, as well as the constructive feedback on the writing. We are committed to improving the clarity and readability of our paper, and we will address the concerns and suggestions raised. Below, we provide detailed responses to your specific points:
>
> For Correctness
>
> Q1 Dataset quality
>
> The reviewer expressed concern about our reliance on non-peer-reviewed methods, specifically Ken Tsui's fastText classifier and the 'Textbooks are all you need' paper. We understand and acknowledge the concern regarding the reliance on works that are still in the preprint stage. However, these achieve significant recognition within the research community. The methodologies discussed in these works align closely with our objectives in evaluating data quality, particularly given that fastText-based classifiers have become a mainstream approach for dataset evaluation and filtering, as demonstrated by their usage in peer-reviewed papers[1, 2]. Unfortunately, many of the classifiers used in these studies have not been made publicly available. Ken Tsui's fastText classifier, being an open-source model, has also garnered  recognition to some extent, making it a reasonable choice for our quality assessment.
>
> Additionally, we will revise Table 3 in our paper by replacing some cited results with those obtained from our independent replication of the experiments, ensuring fairness and transparency in our evaluation. Here is a comparison of our educational value with the dataset cited in Table 3：
>
> | Dataset        | Educational Value (↑) |
> |----------------|-----------------------|
> |PILE          | 1.011                 |
> | OpenWebMath    | 1.089                 |
> | Proof of Pile  | 1.13                  |
> | MATHPILE       | 1.25                  |
> | KNOWLEDGE PILE | 1.29                  |
>
> Q2 Model evaluation
>
> The evaluation of the models, included to show the superiority of the proposed dataset, lacks credible baselines. Indeed, the paper compares models trained on the proposed domain-specific dataset and base models. A comparison with models trained on other datasets (such as the ones in Section 2) is in my opinion necessary.
>
> We understand and acknowledge the reviewer's suggestion to include comparisons with models trained on other datasets. However, training additional baselines is prohibitively expensive. As we noted in our paper, training on 200B tokens using a 7B model would consume 14,400 A800 GPU hours, making multiple baselines impractical.
>
> Recent studies, such as LLaMA, have shown that performance gains in large language models tend to plateau in the later stages of pre-training, indicating that continued training on random subsets offers diminishing returns. To support this, we conducted a set of smaller-scale experiments comparing the performance of models trained from scratch on 200B tokens of KNOWLEDGE PILE and C4 datasets using a 2B LLaMA architecture.
>
> | Dataset        |  MMLU |
> |----------------|:-----:|
> |C4    | 28.52 |
> | KNOWLEDGE PILE | 33.13 |
>
> As shown in this table, KNOWLEDGE PILE outperforms general pretrained dataset by over 5 points, demonstrating its superior performance.
>
> Regarding the comparison with datasets mentioned in Section 2, unlike manual data collection efforts, Query of CC (QoC) requires significantly less manual effort to gather domain-specific data. For example, Pile sources data from 22 different fileds, requiring extensive manual effort to format and initialize them. The OpenWebText dataset involves numerous filters to extract high-quality math-related texts. In contrast, QoC efficiently collects a large volume of high-quality domain-specific data from processed public corpora with minimal manual intervention.
>
> [1] Tom B. Brown, Benjamin Mann and Nick Ryder etc. Language models are few-shot learners. In Hugo Larochelle, Marc’Aurelio Ranzato, Raia Hadsell, Maria-Florina Balcan, and Hsuan-Tien Lin, editors, Advances in Neural Information Processing Systems 33: An-
> nual Conference on Neural Information Processing Systems 2020, NeurIPS 2020, December 6-12, 2020, virtual, 2020. URL https://proceedings.neurips.cc/paper/2020/hash/1457c0d6bfcb4967418bfb8ac142f64a-Abstract.html.
>
> [2] Alon Albalak, Yanai Elazar and Sang Michael Xie A survey on data selection for language models. Trans.
> Mach. Learn. Res., 2024, 2024.

---

> > ### Author Rebuttal · Authors · 2024-08-16
> >
> > Clarity
> >
> > > Q1 The manuscript is compiled with the preprint option, which removes line numbers from the output file.
> >
> > Our manuscript was compiled with the preprint option in accordance with conference requirements, which explains the absence of line numbers. Including the following in LaTeX template:
> > ```
> > % to compile a preprint version, add the [preprint] option, e.g.:
> > %     \usepackage[preprint]{neurips_data_2024}
> > % This will indicate that the work is currently under review.
> > ```
> >
> > > Q2 A great number of references are to preprints available only on ArXiv.
> >
> > > Q3 The references include links to GitHub and Hugging Face, which should instead be clearly marked as footnotes.
> >
> > We chose to reference ArXiv preprints because these papers are highly influential and represent the latest developments in our research area. However, we recognize the importance of citing peer-reviewed work and will review and replace preprint references with formally published literature wherever possible.
> >
> > Regarding the inclusion of GitHub and Hugging Face links, we agree with your suggestion that these should be more clearly marked as footnotes. We will revise the manuscript to follow this format, ensuring a more standardized presentation.
> >
> > > Q4 There are multiple instances of words used incorrectly, inconsistent capitalization, article usage, and incorrect punctuation.
> >
> > > Q5 There are non-standard and unclear expressions that are not explained. For instance, "data recipe", "highly scattered domain data distribution", "queries around the seed keywords in both depth and breadth", "employing LLMs to generate the cognitive processes necessary for answering questions".
> >
> > > Q6 Text styles are often used in unconventional ways. The title contains words in italics, bold text is used for emphasis (instead of italics), and "Knowledge Pile" is (almost) always in teletype (\texttt). This makes the paper more difficult to read than it should be.
> >
> > We will rigorously review the manuscript to address the language and expression issues you highlighted, particularly those related to incorrect word usage, inconsistent capitalization, article usage, and punctuation errors. We will thoroughly review and revise the manuscript to ensure that the language is consistent and precise.
> >
> > For non-standard or unclear expressions, such as "data recipe," "highly scattered domain data distribution," "queries around the seed keywords in both depth and breadth," and "employing LLMs to generate the cognitive processes necessary for answering questions," we understand that these terms may not be clear or standardized. We will revise these sections and either replace these terms with more accurate, widely accepted terminology or provide additional explanations to enhance clarity.
> >
> > > Q7 Figure 4 (left): this figure, with indices of an array on the x-axis, is unnecessarily hard to parse; consider a single plot with logarithmic scale y-axis.
> >
> > Figure 4 (left) shows the distribution of data across different frequency intervals. The x-axis represents the data divided into four intervals by frequency, and the y-axis shows the counts within each interval. The figure illustrates that as frequency increases, the data distribution becomes more skewed, with the highest frequency interval containing 100 web domains that appear far more frequently than others, leading to a significant imbalance in distribution. We will address your suggestion to use a logarithmic scale on the y-axis in future revisions to improve readability.
> >
> > > Q8 Figure 4 (right): why are the bars of different colors if there is no hue?
> >
> > The different colors in Figure 4 (right) represent different time periods, intended to make the data representation clearer. We will clarify this in the figure caption to ensure that the reader understands the purpose of color differentiation.
> >
> > Finally Thank you for your constructive comments. I hope the above explanation can solve your problem. If you have any questions, please leave a comment and we will answer you as soon as possible.

---

> > > ### Comment · Reviewer_menD · 2024-08-23
> > >
> > > I thank the authors for their answers, comments, and availability.
> > > I would, as in my previous review, like to point out that I am not sure whether the methodology is novel enough.
> > >
> > > # Dataset quality
> > >
> > > As mentioned in the survey you are citing, FastText has been used to compare a given dataset to a (supposedly) high-quality reference corpus [1, Section 3.3].
> > > In [2], the reference corpus is a set of Python scripts annotated with GPT4.
> > >
> > > Tsui's post on Hugging Face, however, does not discuss the details of how FastText [3] was trained, on what data, and how this data was gathered.
> > > Thus, even though the model is openly available, it's unclear exactly what notion of 'educational value' it has learned.
> > >
> > > If the authors measure the quality of their dataset with this metric, they should motivate their choice and make the reader aware that there are alternatives.
> > >
> > > Moreover, on closer inspection of Tsui's post, there are multiple synthetic datasets achieving an educational value of above 1.6 (compared to Knowledge Pile's 1.29).
> > >
> > > To summarize my points:
> > >
> > > 1. If 'educational quality' is a valid metric to evaluate datasets, what makes your dataset better than the synthetic ones discussed on Ken Tsui's post?
> > > 2. If not, please use another metric.
> > > 3. Existing, alternative, metrics should be discussed.
> > >
> > > # Clarity
> > >
> > > **Q1** **Submission format**
> > >
> > > From the NeurIPS submission guidelines (`neurips_data_2024.pdf`, lines 31--32):
> > >
> > > > At submission time, please omit the final and preprint options. This will add line numbers to
> > > >  aid review.
> > >
> > > **Q7 Figure**
> > >
> > > Please stick to a conventional way to plot it. In particular, instead of partitioning the x-axis in 4 arbitrary intervals, consider using one lineplot or a scatterplot of count VS frequency.
> > > Furthermore, the caption of Figure 4 should clearly explain what "count" and "frequency" refer to (frequency of ..., count of ...).
> > >
> > > # Documentation
> > >
> > > This part, despite highlighting a violation of the submission guidelines, has not been addressed.
> > >
> > >
> > >
> > > ---
> > >
> > > [1] Albalak, Alon, et al. "A survey on data selection for language models." *arXiv preprint arXiv:2402.16827* (2024).
> > >
> > > [2] Gunasekar, Suriya, et al. "Textbooks are all you need." *arXiv preprint arXiv:2306.11644* (2023).
> > >
> > > [3] Joulin, Armand, et al. "Fasttext. zip: Compressing text classification models." *arXiv preprint arXiv:1612.03651* (2016).

---

> > ### Author Rebuttal · Authors · 2024-08-28
> >
> > Thank you for your comment,
> >
> > **For Dataset quality**
> >
> > We believe that the existing "Education Quality" indicators can reflect the quality of the dataset to a certain extent. However, we contend that directly comparing natural data with synthetic data is not entirely fair for the following reasons:
> >
> > 1. Synthetic data is often meticulously designed for specific purposes, potentially leading to better performance in certain metrics, such as educational quality, this design bias may result in artificially higher scores for those metrics. However, the construction of synthetic data is relatively costly, requiring large models for inference, and it tends to exhibit lower authenticity and diversity compared to real data[1]. Additionally, synthetic data introduces challenges such as hallucinations and risks of training collapse[2].
> >
> > 2. In contrast, real data is more diverse and authentic. Although it may not perform as well on some predefined indicators compared to synthetic data, it better reflects the complex real-world scenarios. We maintain that real data has irreplaceable value in many application contexts.
> >
> > Additionally, we believe that discussing other existing metrics is necessary. In Section 4, we continued pretraining LLaMA and Mistral based on the Query of CC, and compared the performance differences between the original models and the further trained ones on downstream tasks. We are also developing and training similar quality classifiers that will enable us to more accurately assess and improve dataset quality.
> >
> > **For Q1 Submission format**
> >
> > Thank you for reminding us of the submission format requirements. We only noticed the prompts at compilation and overlooked the options for "final" and "preprint." We apologize for any confusion this has caused.
> >
> > **For Q7 Figure**
> >
> > We appreciate your suggestions for improving the images. We have noted your comments on Figure 4 and will adopt a more traditional representation to present the data more clearly.
> >
> > **Document Acquisition and Dataset Access:**
> >
> > Initially, we considered implementing a simple application process for dataset access, given the potential risks associated with large datasets. However, in light of your feedback, we have decided to make the dataset directly available.
> >
> > Thank you for highlighting the issue of missing dataset documentation. This may have been an oversight during the preparation of our submission, and I sincerely apologize for this lapse. To rectify this, I will take the following immediate actions:
> >
> > - Dataset Documentation and Intended Use: I will prepare a comprehensive dataset document that includes data descriptions, intended uses, and a recommended documentation framework. This framework will encompass data tables, data nutrition labels, NLP data statements, data cards, and accountability frameworks.
> >
> > - Author Statement: I will provide a statement clarifying the dataset's copyright, privacy, usage rights, and responsibilities, confirming data licensing.
> >
> > - Hosting, Licensing, and Maintenance Plans: I will detail the dataset’s hosting platform, licensing agreement, and maintenance plans to ensure its accessibility and long-term availability.
> >
> > Thank you once again for your invaluable feedback.
> >
> > [1] Liu, Ruibo, et al. "Best practices and lessons learned on synthetic data for language models." arXiv preprint arXiv:2404.07503 (2024).
> >
> > [2]Shumailov, I., Shumaylov, Z., Zhao, Y., Gal, Y., Papernot, N., & Anderson, R. (2023). The Curse of Recursion: Training on Generated Data Makes Models Forget. ArXiv, abs/2305.17493.

---

> > > ### Comment · Reviewer_menD · 2024-08-29
> > >
> > > Thank you for your replies. As you promise that the necessary documentation will be available, I have updated my score accordingly.

---

### Author Rebuttal · Authors · 2024-08-16

We sincerely appreciate the reviewers' recognition of the novelty and potential of our idea. Expanding domain-specific queries using large language models to retrieve and collect high-quality data in specific fields is promising. Most of the criticisms focused on the lack of baselines and the clarity of our descriptions, and many of these critiques were constructive. In addressing these concerns, we further explored specific aspects of the proposed technique, which significantly strengthened the arguments presented in our paper. We would like to clarify the following points regarding the common issues raised by the reviewers:

**Regarding Baselines**:  As mentioned in the paper, training Llama-Qoc (7B) on the Knowledge Pile requires approximately 14,400 A800 GPU hours. Due to cost constraints, we were unable to introduce additional baselines, such as datasets generated by random sampling from Common Crawl (CC). Some studies have shown that performance improvements of large language models typically plateau during the later stages of pre-training, indicating that the performance gains from randomly sampled CC data diminish over time. However, our dataset demonstrated significant performance improvements in the later stages of training, which underscores the high quality of our dataset to some extent. Additionally, we attempted to train a 1.8B parameter LLaMA-architecture model from scratch, comparing with Knowledge pile and a randomly sampled 200B subset of C4[1] (which is a general pretrained dataset from Common Crawl) as a baseline:

| Dataset        |  MMLU |
|----------------|:-----:|
|C4    | 28.52 |
| Knowledge Pile | 33.13 |

This comparison highlights the performance gains of the Knowledge Pile over randomly sampled data.

**Quality evaluation**: We compared the educational value of the domain-specific datasets included in Table 1:

| Dataset        | Educational Value (↑) |
|----------------|-----------------------|
|PILE          | 1.011                 |
| OpenWebMath    | 1.089                 |
| Proof of Pile  | 1.13                  |
| MATHPILE       | 1.25                  |
| KNOWLEDGE PILE | 1.29                  |

As shown in the table, the Knowledge Pile demonstrates a clear advantage in quality compared to other datasets.
We greatly appreciate all reviewers' comments, and we would be grateful for any further feedback you may provide during the upcoming discussion period.

[1] Colin Raffel, Noam Shazeer, Adam Roberts, Katherine Lee, Sharan Narang, Michael Matena, Yanqi
Zhou, Wei Li, and Peter J. Liu. Exploring the limits of transfer learning with a unified text-to-text
transformer. J. Mach. Learn. Res., 21:140:1–140:67, 2020

---

### Decision · Program_Chairs · 2024-09-26

**Decision:**

Reject

**Comment:**

This work describes both an approach to automatically construct domain specific corpora from Common Crawl using LLMs. It's an interesting approach where the LLM is used to generate domain specific queries which are then used to retrieve document subsets. The dataset released is based on the selection in several domains and looks ready to use and is well documented. The paper demonstrates that the domain specific corpora gathered in this fashion looks to have better properties around knowledge and improves performance. The approach to collection and dataset look quite useful.

The paper could have done a better job in some places connecting their documentation together. However, the authors addressed this effectively in the discussion and I think it's all addressed. While the paper analyses the quality of the dataset and the impact of errors that could be produced by the LLM for retrieval in a systematic way, the authors could have done a bit better job reflecting on the implications of the datasets they have produced from common crawl.

One reviewer did raise the issue of use of arXiv papers and non-peer reviewed work. The authors addressed this effectively. Overall, this paper is close. I think the fundamentals are all there and the clarifications have/can be made. Hence, I'm recommending an accept.

Note from PC: This year, the track has been incredibly competitive, which meant that many good papers had to be rejected. After careful discussion with the SACs we have concluded that this paper unfortunately cannot be accepted this time. This is the final decision, which cannot be appealed. We encourage the authors to incorporate feedback from reviewers and additional results / discussion provided during the author response period in their next submission.